# Acoustic duetting in *Drosophila virilis* relies on the integration of auditory and tactile signals

Kelly M LaRue[1,2], Jan Clemens[1,2], Gordon J Berman[3], Mala Murthy[1,2]*

[1]Princeton Neuroscience Institute, Princeton University, Princeton, United States; [2]Department of Molecular Biology, Princeton University, Princeton, United States; [3]Lewis Sigler Institute for Integrative Genomics, Princeton University, Princeton, United States

**Abstract** Many animal species, including insects, are capable of acoustic duetting, a complex social behavior in which males and females tightly control the rate and timing of their courtship song syllables relative to each other. The mechanisms underlying duetting remain largely unknown across model systems. Most studies of duetting focus exclusively on acoustic interactions, but the use of multisensory cues should aid in coordinating behavior between individuals. To test this hypothesis, we develop *Drosophila virilis* as a new model for studies of duetting. By combining sensory manipulations, quantitative behavioral assays, and statistical modeling, we show that *virilis* females combine precisely timed auditory and tactile cues to drive song production and duetting. Tactile cues delivered to the abdomen and genitalia play the larger role in females, as even headless females continue to coordinate song production with courting males. These data, therefore, reveal a novel, non-acoustic, mechanism for acoustic duetting. Finally, our results indicate that female-duetting circuits are not sexually differentiated, as males can also produce 'female-like' duets in a context-dependent manner.

*For correspondence: mmurthy@princeton.edu

**Competing interests:** The authors declare that no competing interests exist.

## Introduction

Studies of acoustic communication focus on the production of acoustic signals by males and the arbitration of mating decisions by females. However, for many species of primates (*Haimoff, 1986*), birds (*Hall, 2004*), frogs (*Tobias et al., 1998*), and insects (*Bailey, 2003*), females also produce songs, and duets are common; moreover, recent studies suggest that female song production may be ancestral (*Wiens, 2001*; *Odom et al., 2014*). Animal duets involve predictable response latencies between the calls of males and females; that is, males and females do not sing simultaneously, as human duetters do, but rather interchange acoustic signals (*Bailey and Hammond, 2003*). Duetting species can be grouped into two classes: those that answer their partner's song without fine-scale coordination of song syllables or elements (polyphonal duetters) and those that synchronize within a song bout (antiphonal duetters) (*Hall, 2009*). Regardless of duet type, each individual must adjust the rate and timing of his/her courtship songs relative to each other. Therefore, duetting requires speed and accuracy in both detection of a partner's signal and the production of a response. Latencies between male and female songs are reported to be as short as tens of milliseconds for some birds (*Logue et al., 2008*; *Fortune et al., 2011*) and insects (*Heller and von Helversen, 1986*; *Rheinlaender et al., 1986*). For example, the antiphonal duets of plain-tailed wrens are so rapid that they sound as if produced by a single animal (*Mann et al., 2006*). Both male and female wrens display differences in inter-syllable intervals when singing alone vs with a partner, indicating that sensory perception plays an ongoing role in shaping song timing (*Fortune et al., 2011*). In support of

**eLife digest** When performing a duet, human singers listen to each other and also use other cues like tapping and nodding to keep in time. Duets are also found in the courtship rituals of some animals, but unlike human duetters—who tend to sing at the same time—the male and female animals take turns singing back and forth in quick succession. For most species, the male starts the duet and the female responds. It is thought that animals rely on their sense of hearing to perform their duets successfully, but it is not clear whether they also rely on other cues, or what these might be.

Fruit flies produce songs during courtship displays by vibrating their wings. Typically, only the male fly in a pair will sing, while the female is silent and makes the mating decisions. However, both the male and female sing in the courtship rituals of a fruit fly species called *Drosophila virilis*. Now, LaRue et al. have studied the songs and other behaviors produced by this fruit fly during courtship displays.

The experiments show that *D. virilis* males and females perform duets; in other words, there is a predictable timing between male and female songs. For the female fly to produce the right amount of song, she needs to detect sound cues from the male fly and also be in physical contact with him. This contact involves the male tapping the female's abdomen and licking her genitalia, and the precise timing of these behaviors predicts when she sings. This behavior was also observed in duets produced by male-only pairs, so it is not gender-specific.

These findings show that *D. virilis* flies use both sound and other types of cues to coordinate singing duets during courtship. LaRue et al. suggest that female flies may choose to mate with males that provide multiple timing cues during the duet. A future challenge is to understand how information provided by the different cues is combined in the brain of the female to drive this courtship behavior.

this, recordings from pre-motor areas of the wren brain (HVC) reveal auditory responses that are tuned to duets (*Fortune et al., 2011*); that is, neural centers that drive song production also directly integrate auditory information. For polyphonal duetting bushcrickets, which like many insects produce sounds via their wings, auditory information is transmitted not only to the brain but also directly to wing-controlling neural ganglia; thus, similar to birds, this suggests that song pattern-generating circuits (housed in these ganglia) directly integrate auditory information to shape duets (*Rheinlaender et al., 1986*).

While auditory perception appears to play an important role in producing duets, no studies to date have investigated a role for non-acoustic cues. However, for many other behaviors, the use of multisensory cues is proposed to improve signal detection (*Rowe, 1999*; *Mirjany et al., 2011*; *McMeniman et al., 2014*), synchronization (*Elliott et al., 2010*), or decision-making abilities (*Kulahci et al., 2008*; *Raposo et al., 2012*). For example, mosquitos locating human hosts rely on both thermal and olfactory cues; this dual requirement increases the fidelity of host-seeking (*McMeniman et al., 2014*). Similarly, bees have been shown to make faster and more accurate-foraging decisions when choosing between flowers that differ in multiple modalities (vs a single modality) (*Kulahci et al., 2008*). These studies (and others) lend support to the idea that acoustic-duetting behaviors, which are under tight temporal control, might also benefit from the use of multimodal sensory processing. Moreover, because many species produce songs that are not stereotyped (*Kovach et al., 2014*), the use of multiple sensory cues should aid in response timing.

To test this hypothesis, we developed *Drosophila virilis* as a new model system for the study of acoustic coordination. Song production (via wing vibration) is an integral component of the courtship ritual among Drosophilid flies (*Ewing and Bennet-Clark, 1968*), but typically, as in *Drosophila melanogaster*, only males produce song (*Dickson, 2008*). The presence of female song has been reported in only a handful of *Drosophila* species (*Donegan and Ewing, 1980*), and it is not known if any of these species duet. Here, by combining a battery of specific sensory manipulations, quantification of a large number of courtship songs, and statistical modeling, we not only show that *D. virilis* males and females duet but also uncover the underlying mechanisms. We find that females integrate both auditory (male song) and tactile (male contact with her abdomen and genitalia) cues for

song production and coordination. Moreover, the precise timing of male tactile cues predicts female song timing, revealing a novel, non-acoustic, mechanism for acoustic coordination. Finally, we also address the importance of male and female song production for courtship success, and demonstrate, by comparing duets produced in male–male pairings, that acoustic-duetting behavior in females is not sexually differentiated.

## Results

### *D. virilis* males and females coordinate song production into an acoustic duet

Previous studies have documented the presence of female song in the *D. virilis* group of species (*Satokangas et al., 1994*); however, it is not known if males and females coordinate their song production into duets nor have the mechanisms underlying female song production been studied. By combining video and audio recordings, we matched *D. virilis* male and female wing movements with acoustic signals to accurately classify male and female song (*Figure 1A–C* and *Video 1*). We recorded song in a multi-channel recording apparatus (*Arthur et al., 2013*). We chose *virilis* strain 15010–1051.47, because males and females of this strain sang robustly in our courtship chambers (*Figure 1—figure supplement 1*). Consistent with previous reports (*Huttunen et al., 2008*), we found that *virilis* males (when paired with a virgin female) produce highly stereotyped bouts of pulses via unilateral wing vibration (*Figure 1A,D*). We define a bout as a stretch of song from either a male or female, which is separated from the next stretch of song by more than 150 ms. Each male bout contains $6.9 \pm 1.2$ pulses with an inter-pulse interval (IPI) of $21.2 \pm 1.9$ ms (*Figure 1E*). Females (when paired with a virgin male), in contrast, use bilateral wing vibration to generate variable-length bouts of pulses separated by longer (and more variable) IPIs ($7.2 \pm 6.2$ pulses per bout, with IPIs of $55.2 \pm 26.3$ ms) (*Figure 1—figure supplement 2* and *Figure 1E*). The frequency spectra of individual male pulses show a peak at higher frequencies relative to individual female pulses (*Figure 1F*). Although there are instances of song overlap between males and females, this behavior is extremely rare (*Figure 1C* and *Figure 1—figure supplement 3E* [population median of overlaps = 0%]). Based on these characteristics, we modified our software for automated segmentation of *D. melanogaster* song (*Arthur et al., 2013*) to segment *virilis* male and female pulses (*Figure 1—figure supplement 3*).

Acoustic duets are characterized by predictable response latencies between male and female calls (*Bailey, 2003*). To determine if *virilis* males and females duet during their courtship ritual, we characterized response times between male bouts (which are highly stereotyped, see *Figure 1*) and female pulses and vice versa. The distribution of female response times (relative to the onset of a male bout) is peaked at 409 ms and is significantly different from the distribution of response times calculated from randomized versions of the data set in which either male bout times or female pulse times were shuffled (*Figure 2A* and see 'Materials and methods'). Thus, female song is temporally coordinated with male song. Male response times to female song have not been examined previously in any insect. We calculated the delay between the onset of male bouts and the center of each pulse in the preceding female bout (*Figure 2—figure supplement 1*). Male response times to the first, second, penultimate, and last pulse in a female bout were all significantly different from response times from randomized data, with responses to the last pulse showing the largest difference (*Figure 2—figure supplement 1C*). We, therefore, defined male response times as the latency between the onset of the male bout and the previous female pulse (*Figure 2B*) and found that the distribution of male response times is peaked at short delays (110 ms), compared with female response times (even when accounting for differences in how response times are calculated for males and females). To the best of our knowledge, these data represent the first quantitative evidence of acoustic duetting in a Drosophilid species.

To determine if response timing relies on hearing the song of the partner, we rendered either the male or female deaf by removing the arista, a feathery appendage of the antenna that serves as the acoustic receiver (*Gopfert and Robert, 2002*). Intact females maintained their response timing in pairings with arista-cut (AC) males (*Figure 2C*, peak at 407 ms); however, song bouts from AC males no longer followed female pulses with a predictable response latency (*Figure 2D*). Instead, AC male response times were not significantly different from response times from randomized versions of the data (p = 0.24). This effect was specific to response timing, because AC males maintained wild-type (WT) levels of song production (Figure 4B, manipulation 3), and showed no change in inter-bout

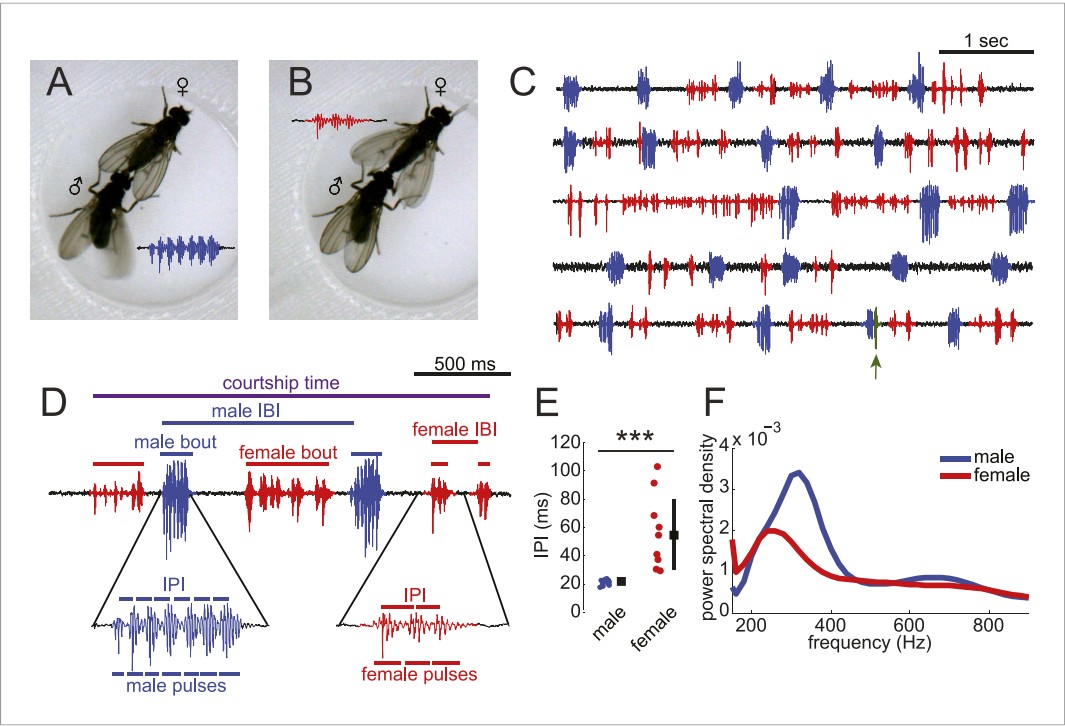

**Figure 1**. *Drosophila virilis* males and females produce distinct courtship songs. Schematic of the wing movement during the production of courtship song in *Drosophila virilis* male (**A**) and female (**B**). (**C**) Examples of acoustic duets between male (blue) and female (red) in five wild-type (WT) courtships. Regions of song overlap (arrow) are shaded in green. (**D**) Detailed view of male (blue) and female (red) song produced during courtship. Song parameters described in the 'Materials and methods' and used in all analyses are indicated. IBI = inter-bout interval and IPI = inter-pulse interval. (**E**) The median (per individual) IPI for male (blue) and female (red) song; male and female pulses were identified by matching acoustic and video recordings (*Video 1*). Black squares show population mean and black bars standard deviation (n = 13 courtships, Student's *t*-test ***p < 0.001). (**F**) Power spectral density for pulses from males (n = 3854, blue) and females (n = 497, red). Based on the differences in IPI and pulse frequency between male and female song, we created a semi-automated software pipeline to segment *virilis* song (*Figure 1—figure supplement 2*).

The following figure supplements are available for figure 1:

**Figure supplement 1**. Assessment of song production in *D. virilis* strains.

**Figure supplement 2**. Bout durations of male and female courtship songs.

**Figure supplement 3**. Development of song segmentation software for *Drosophila virilis*.

interval (IBI) or pulse frequency (*Figure 2—figure supplement 2*). Thus, males rely on hearing their partner's song for acoustic coordination. In contrast, AC females paired with intact males maintained coordination (*Figure 2E*, peak at 408 ms), even though this manipulation caused an up-regulation in song production (Figure 4E, manipulation 3) and a shift in IPI and pulse frequency (*Figure 2—figure supplement 2*). This result suggests that females may use a non-auditory cue to coordinate song production with their partner.

## Testing the importance of male and female song in courtship success

To determine the role of song in mating success, we removed male or female acoustic cues by either amputation of the wings (rendering the fly mute) or the aristae (rendering the fly deaf). Removal of either appendage did not reduce courtship rates (*Figure 3—figure supplement 1*). However, removing either the production of male song (in pairings between WT females and wing-cut (WC)

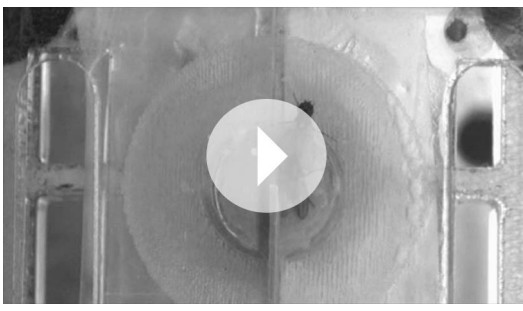

**Video 1.** *D. virilis* duetting behavior. Representative example of courtship between a naive *D. virilis* male and female pair. Shown in the video is one channel/chamber of a multi-channel song recording system. Video is acquired at 15 Hz.

males) or the detection of male song (in pairings between WT males and AC females) dramatically reduced the percentage of copulating pairs (*Figure 3A*). These results indicate that male song is required for mating success, consistent with data from other closely related species (*Aspi and Hoikkala, 1993*). Because females increase song production rates (a parameter distinct from IPI, see 'Materials and methods') in the absence of male song (*Figure 3—figure supplement 2*), increased female song production might cause the observed decrease in copulation rates. Regardless, both interpretations support a role for male song, either directly or indirectly, in mating success.

To test the role of female song in mating, we first examined WT males paired with mute (WC) females and found a significant reduction in copulation (*Figure 3B*, yellow solid vs yellow dashed control). However, deaf males, which cannot detect female song, still copulated at WT rates when paired with WT females (*Figure 3B*, light blue solid vs light blue dashed control). Because *virilis* females indicate their receptivity via wing spreading

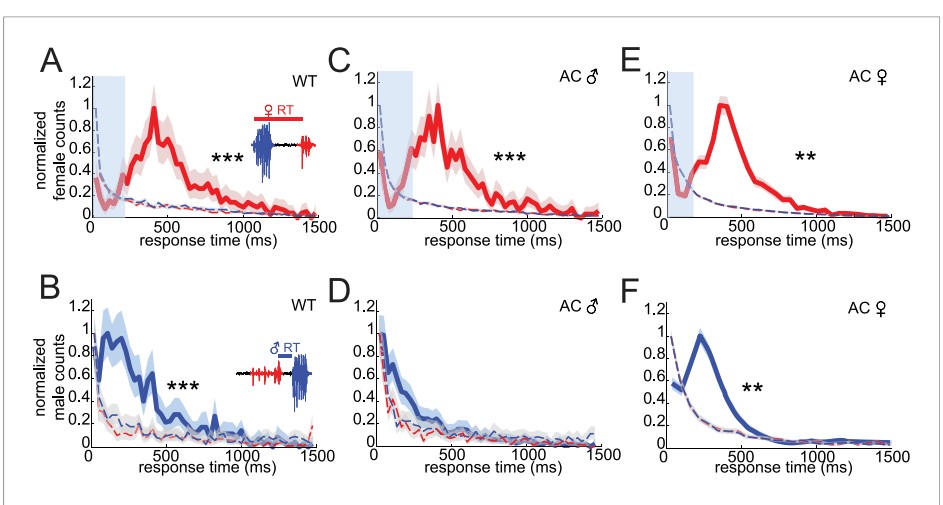

**Figure 2**. *Drosophila virilis* courting pairs coordinate song production into an acoustic duet. (**A**) Female response times were calculated as the time between male bout onset and the following female pulse (inset). (**B**) Male response times were calculated as the time between the onset of the male bout and the preceding female pulse (inset). Normalized distributions of female (red, **A**, **C**, and **E**) and male (blue, **B**, **D**, and **F**) response times for: WT females paired with WT males (**A**, n = 970 response times and **B**, n = 1031 response times, n = 89 courtships), WT females paired with arista-cut (AC, deaf) males (**C**, n = 1173 response times and **D**, n = 1206 response times, n = 124 courtships), and AC (deaf) females paired with WT males (**E**, n = 3391 response times and **F**, n = 3948 response times, n = 34 courtships). Response times were also calculated after shuffling either male bout (blue dashed) or female pulse (red dashed) intervals. Comparisons with these null distributions reveal that only removing male hearing impacts response timing (Kolmogorov–Smirnov (K-S) Test ***p < 0.001 **p < 0.01). Basic song structure remains stable for the AC manipulations (*Figure 1—figure supplement 2*). Transparent blue bar (**A**, **C**, and **E**) indicates average male bout duration. Shading around each response time distribution represents bootstrapped 95% confidence intervals.

The following figure supplements are available for figure 2:

**Figure supplement 1**. Quantification of male response times.

**Figure supplement 2**. Song parameters of AC flies.

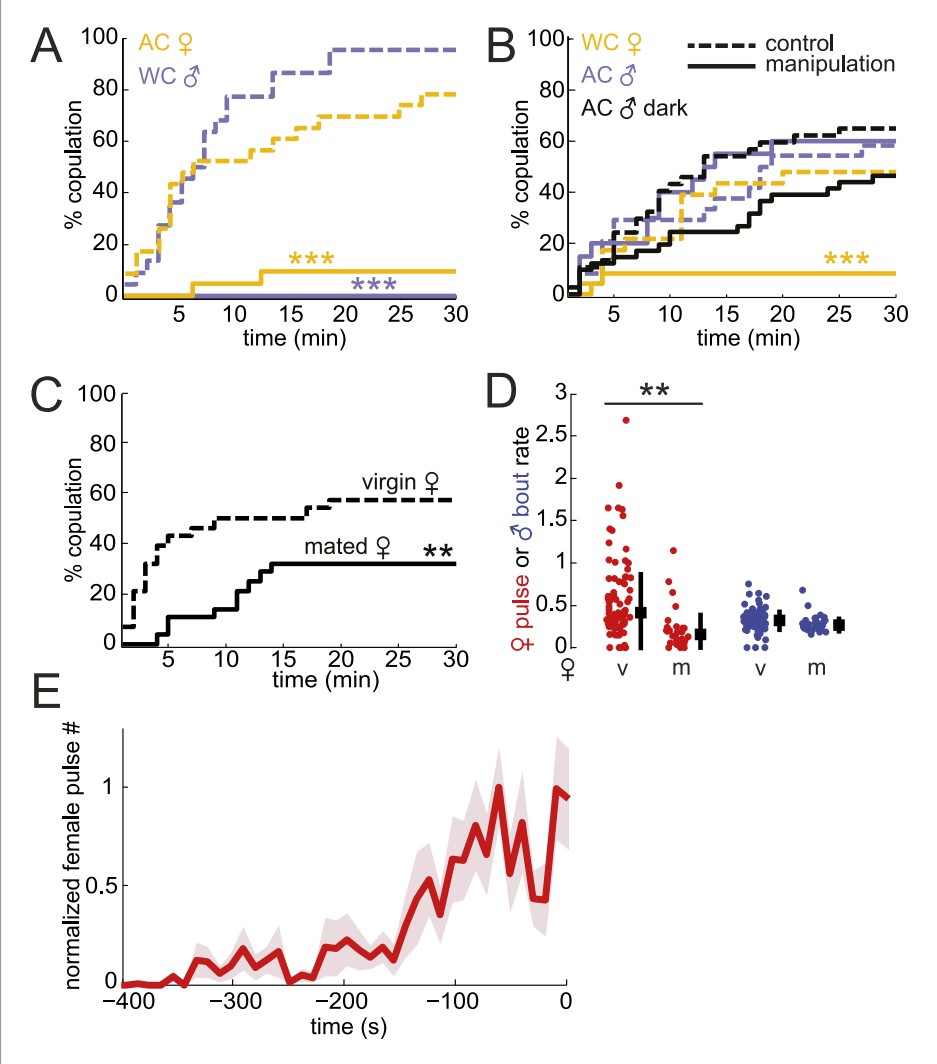

**Figure 3**. Testing the role of male and female song in courtship success. (**A** and **B**) Percent copulation over the 30-min observation period for manipulated flies (WC, wing-cut [mute]; AC, dark, lights off [no visual cues]). Each manipulated fly was paired with a WT fly of the opposite sex (solid lines). Latency to mating was compared to WT controls prepared under similar conditions (dashed lines). The numbers of individuals in each experiment were: WC male n = 20, WC male control n = 22, AC female n = 21, AC female control n = 23, WC female n = 24, WC female control n = 23, AC male n = 20, AC male control n = 24, AC male dark n = 41, AC male control dark n = 37 (Mantel–Cox test, ***p < 0.001). Experimental manipulations did not affect courtship rates (*Figure 3—figure supplement 1*). (**C**) Percent copulation over the 30-min observation period for mated females paired with virgin males (solid black, n = 28) vs virgin females paired with virgin males (dashed black, n = 28, Mantel–Cox test, **p < 0.01). (**D**) Median female pulse (red) or male bout (blue) rate (see 'Materials and methods') dependent on whether the female is virgin (v, n = 83) or mated (m, n = 26), independent of copulation success (*Figure 3—figure supplement 2*). Black squares are the population median and black bars the interquartile range (IQR) (generalized linear model [GLM] t statistic **p < 0.01). (**E**) Mean female pulse number (normalized) in 10-s bins prior to the moment of copulation (n = 8 courtships). Shading represents s.e.m.

The following figure supplements are available for figure 3:

**Figure supplement 1**. Courtship activity of manipulated flies in latency to mating assays.

**Figure supplement 2**. Female pulse rates in the absence of male acoustic cues.

**Figure supplement 3**. Song rates in successful versus unsuccessful courtships.

(*Vuoristo et al., 1996*), deaf males may use this visual signal (produced by intact females) to drive mating in the absence of acoustic cues. To prevent males from observing female wing spreading, we allowed deaf males to court WT females in the dark and subsequently observed a 29% reduction in copulation (*Figure 3B*, black solid vs black dashed control, p = 0.08). The lack of significance of the reduction in courtship success may be due to the ability of males to still detect female wing spreading in the dark (e.g., using mechanosensation). Because AC males do not coordinate their song production with females (*Figure 2D*) but still maintain high-copulation success (*Figure 3B*), our data also suggest that females don't require coordinated duetting for mating when paired with a single male. Duetting, however, may be important for mate selection in the context of competition as has been suggested for other insects and some species of birds (*Bailey, 2003*; *Hall, 2004*).

A role for female song in mating would nonetheless be indicated by a correlation between female song production and her receptivity state. To test this hypothesis, we compared courtship success and song production rates (see 'Materials and methods') of virgin females and mated females. Mated *D. melanogaster* females show a reduction in copulation rates following mating; this is due to the effects of a sex peptide transferred from males to females during copulation (*Villella and Hall, 2008*; *Yapici et al., 2008*). Similarly, we found that mated *virilis* females show significantly lower copulation rates (*Figure 3C*). We, therefore, investigated the association between song production and state of the female. Both virgin *virilis* males and females sang more if they copulated within the 30-min experimental window vs if they did not copulate (*Figure 3—figure supplement 3*). After correcting for this difference (see 'Materials and methods'), we still observed a significant reduction in song production in mated *virilis* females compared with virgin females (*Figure 3D*); this reduction was specific to female song, as males who courted mated vs virgin females produced similar amounts of song. Finally, we observed that *virilis* females, on average, increased song production over the 400 s leading to copulation (*Figure 3E*). These data collectively argue that female song production (and therefore duetting) is not required for mating, but rather is a positive cue that promotes mating.

## Mapping the sensory inputs to *D. virilis* song production pathways

In order to duet, both males and females must regulate their song production rates relative to each other. We next examined the sensory stimuli and pathways that influence male bout and female pulse rates in *D. virilis*. We examined the rate of male bouts and female pulses, because male bouts are highly stereotyped while female bouts are not (*Figure 1—figure supplement 2*); see 'Materials and methods' for a description of how song rates were calculated. We manipulated sensory structures relevant for courtship (*Figure 4A* and see 'Materials and methods') and used statistical modeling to predict song rates from data sets containing all possible combinations of sensory organ manipulations to the male or female (each paired with a WT partner). Based on the interactions we observed in videos of *virilis* courtship (*Video 1*), we manipulated the following sensory structures: (i) aristae (to block the detection of acoustic signals [*Gopfert and Robert, 2002*]), (ii) antennae (to block the detection of volatile pheromones [*Benton, 2007*]), (iii) tarsi (to block the detection of contact pheromones [*Thistle et al., 2012*]), and (iv) female genitalia (to block male contact with the lower abdomen, which is known to be important for *D. virilis* courtship [*Vuoristo et al., 1996*]; see 'Materials and methods'). To block visual cues, we placed flies in the dark. We used generalized linear models (GLMs) to predict male or female song rates based on the sensory manipulations (see 'Materials and methods'). The GLM weights or coefficients reveal the contribution of each sensory channel to song production. We first fit the GLM to data from pairings between manipulated males and WT females and found that bilateral removal of the male tarsi was the only significant predictor of male bout rate (*Figure 4B* and *Figure 4—figure supplement 1*). This result contrasts with results from *D. melanogaster*, where tarsal removal enhances courtship activity in males (*Fan et al., 2013*). We also found that sensory organ manipulation of males has similar effects on female song production (*Figure 4C* and *Figure 4—figure supplement 1*), suggesting that either female song production is dependent on the presence of male song or that some male manipulations also impact sensory perception in females (for example, removal of male tarsi may block a mechanosensory cue to the female; see below).

We next performed a similar panel of sensory manipulations on the female. We found that none of the female manipulations (either alone or in combination) impacted male song production rates (*Figure 4D*); consequently none of the GLM coefficients were significantly different from zero

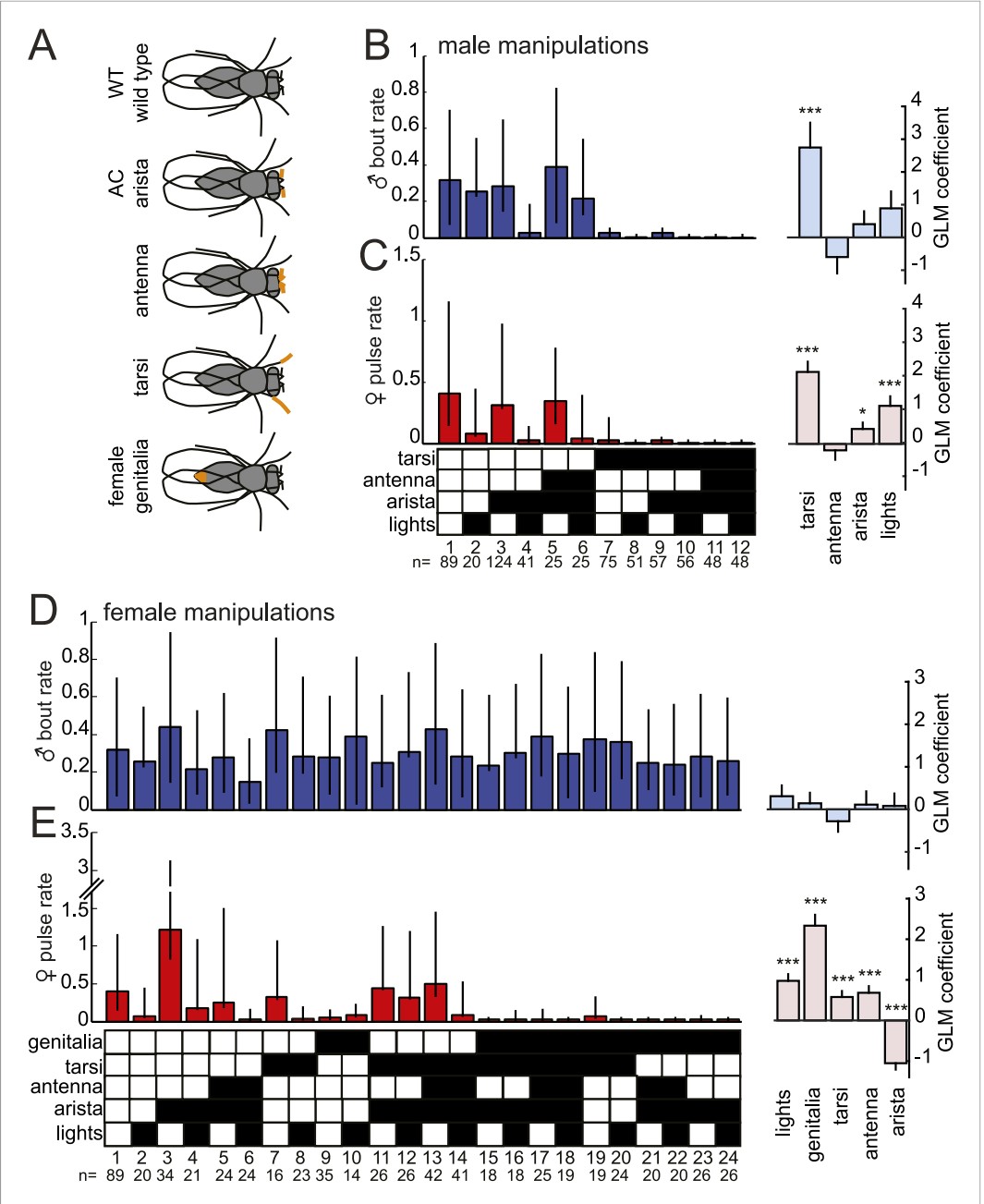

**Figure 4**. Multiple sensory channels influence song production rates in *virilis* males and females. (**A**) Schematic of sensory organ manipulations. Orange shading indicates the region of organ modification (amputation or blocking). Male bout (blue, **B**) and female pulse (red, **C**) rate in pairings between sensory organ manipulated males and WT females. Sensory organ manipulation for each experiment is indicated with a black box. Each colored bar represents the median bout (blue, **B**) or pulse rate (red, **C**) and error bars the IQR. The number of pairs examined is listed below the plots. A GLM was used to predict the song rate based on the presence/absence of different sensory channels. GLM coefficients indicate the relative importance of each sensory channel in determining male song production (light blue) and female song production (pink); error bars represent bootstrap estimate of the s.e.m. (***p < 0.001, *p < 0.05). Similarly, male bout (blue, **D**) and female pulse (red, **E**) rate, along with GLM coefficients (light blue and light pink, respectively), for pairings between WT males and sensory organ manipulated females.

The following figure supplement is available for figure 4:

**Figure supplement 1**. GLM statistics for sensory organ manipulations.

(*Figure 4—figure supplement 1*, p = 0.68). Thus, male song production is unaffected by the state of the female. However, our GLM analysis revealed that all of the evaluated female sensory manipulations impacted female pulse rate (*Figure 4E* and *Figure 4—figure supplement 1*). Interestingly, inputs to the female aristae and genitalia have the largest effects on female song production, but act in opposite directions (*Figure 4E*). Removal of the aristae up-regulates female pulse rate (as does removing male song [*Figure 3—figure supplement 2*]), while blocking the female genitalia down-regulates pulse rate. These results establish a role for multisensory cues in female song rate and indicate that information sensed via the female genitalia and aristae is combined to fine-tune how much song the female produces. We, therefore, next focused on these two sensory pathways to test their role in regulating female song timing relative to her partner.

## D. virilis females rely on non-acoustic cues to coordinate song timing with males

Data from AC females indicated that male auditory cues are not necessary for female song response timing (*Figure 2E*). To further investigate the role of auditory cues in duetting, we examined female acoustic coordination with playback of recorded male song (see 'Materials and methods'). Previous studies of the *D. virilis* species group showed that females respond to the playback of male song by spreading their wings (*Ritchie et al., 1998*; *Isoherranen et al., 1999*), but these studies did not characterize female song production. We were unable to detect any female song in response to either playback alone or to playback in the presence of a decapitated male that does not interact with the female (data not shown). However, females produced abundant song in the presence of a WC male (*Figure 5A*). For these pairings, we calculated female response times relative to the playback stimulus and found the distribution to be significantly different from response times from shuffled data (*Figure 5B*, p < 0.001). However, the distribution of female response times to playback does not resemble the WT distribution (compare with *Figure 2A*). Due to issues (in only the playback experiments) with identifying female pulses that overlap with the male playback (see 'Materials and methods'), we also calculated female response times from the end of the male bout (*Figure 5B* inset). We found that this distribution more closely matched the distribution of response times from shuffled data (although they are still significantly different, p < 0.05). Additionally, we performed the same experiment with AC females and saw similar trends in response time curve shape and significance values (*Figure 5C*). These results, therefore, suggest that auditory cues, when uncoupled with the male behavior, have little influence on female response timing (vs their influence on female song rate [*Figure 4*]). Any relationship between female song and the acoustic playback must be due to indirect effects on the WC male. We, therefore, hypothesized that another cue provided by the WC male influenced female song timing. To test the role of contact cues provided by the male, we next examined pairings between intact males and headless females.

Previous studies showed that headless *D. virilis* females reject male courtship attempts (*Spieth, 1966*), so we were surprised to observe that these females continued to produce song (*Figure 6A*). Furthermore, we observed song production from headless females only in the presence of a male and while he was courting (*Video 2*). Headless females produced pulses with longer IPI but similar frequency spectra, relative to intact females (compare *Figure 6B,C* with *Figure 1E,F*). In addition, headless females produced as much song as intact females; song production rates were significantly reduced only when we additionally blocked the genitalia of headless females (*Figure 6D*). Importantly, headless females still coordinated acoustic signal production with their male partner (*Figure 6E*), although the peak of the response time distribution was broader (more variable) and shifted to the right (longer latency) relative to the intact female distribution (compare with *Figure 2A*).

From these data, we conclude that the neural circuitry that controls female duetting behavior lies largely within the ventral nerve cord and does not require descending activation signals from the brain. However, while male contact with the female genitalia is sufficient to drive the female song circuit, inputs from the brain contribute to response timing, IPI, and the decision to mate (in our experiments, headless females did not perform the characteristic wing-spreading behavior indicative of female acceptance and never copulated with WT males [data not shown]). While song production in decapitated *D. melanogaster* males has been observed (*Clyne and Miesenbock, 2008*), it was elicited only upon optogenetic neural activation of song circuits. The song produced by headless *D. virilis* females, in contrast, does not require experimental stimulation, but simply the natural sensory cues provided by the male partner.

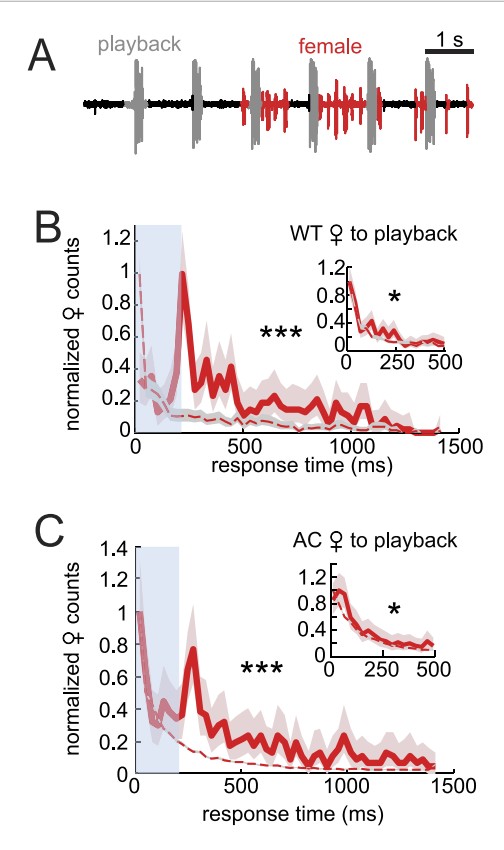

**Figure 5**. Testing the sufficiency of auditory cues for female song production and timing. (**A**) Female pulses (red) produced in response to playback of male song (gray) and in the presence of a WC male. Normalized WT (**B**, red, n = 451, n = 13 courtships) and AC (**C**, red, n = 522, n = 16 courtships) female response time distributions to playback compared with response time distributions from shuffled data (red dashed). Inset: the same data, but female response times are calculated from the end of the male bout, therefore, ignoring overlaps (K-S test ***p < 0.001, *p < 0.05).

## The precise timing of male contact with the female abdomen predicts female song

In contrast with *D. melanogaster*, *D. virilis* males continually contact the female during courtship (*Vedenina et al., 2013*); however, the temporal pattern of male contact with the female has not been characterized. From higher speed videos taken of the ventral side of the courting flies (*Video 3*), we observed that males rub their tarsi on ventral tergites A3–A5 of the female abdomen while also licking (with their extended proboscis) the female genitalia (*Figure 7A*). We, therefore, quantified the temporal dynamics of these motor programs (see 'Materials and methods' and *Figure 7B*) to determine if either behavior predicted male and female song timing. We again used GLMs to predict female pulse and male bout times from male behaviors. More specifically, we predicted the presence/absence of female or male song based on the temporal pattern of behaviors preceding each time point during courtship interactions. The resulting set of coefficients for each behavior combine to produce temporal filters for predicting song (see 'Materials and methods'). Since these behaviors are correlated in time and with each other, we used a sparse prior on the GLM coefficients that penalize non-predictive filter components (*Coen et al., 2014*).

Looking at individual male features predicting female song, we found that tarsal vibrations and proboscis licking were most predictive of the occurrence of female song (*Figure 7C*). The associated filters had their peaks within 16 milliseconds of female song, indicating low-latency coupling of male cues with female song (*Figure 7D*). Predictability was not substantially enhanced with a two-variable model (*Figure 7—figure supplement 1*). Notably, male song was much less predictive of female song, further suggesting that acoustic cues play a subordinate role in female song timing. Interestingly, we found that the male behaviors predicting female song also predict male song. That is, male tarsal vibration and proboscis licking were most predictive of the presence/absence of male song (*Figure 7E*). However, the associated filters peaked 200 and 50 ms preceding male song, respectively (*Figure 7F*). This implies that male courtship behavior forms a behavioral sequence that starts with tarsal vibration, transitions into proboscis licking, and ends with male song.

These results suggest that both male tarsal contact with the female abdomen and male proboscis contact with the female genitalia are independently predictive of female response timing. Above, we showed that blocking just the female genitalia reduces female song production rates (*Figures 4E, 6D*). Given the GLM results, we next blocked segments A3–A5 of the female abdomen, while leaving the genitalia unblocked, and discovered that this manipulation also significantly reduced female song production (*Figure 7G*). However, additionally removing the aristae restored song production rates to WT levels (compare with *Figure 3—figure supplement 2*), indicating that auditory and abdominal sensory inputs are combined to drive song production

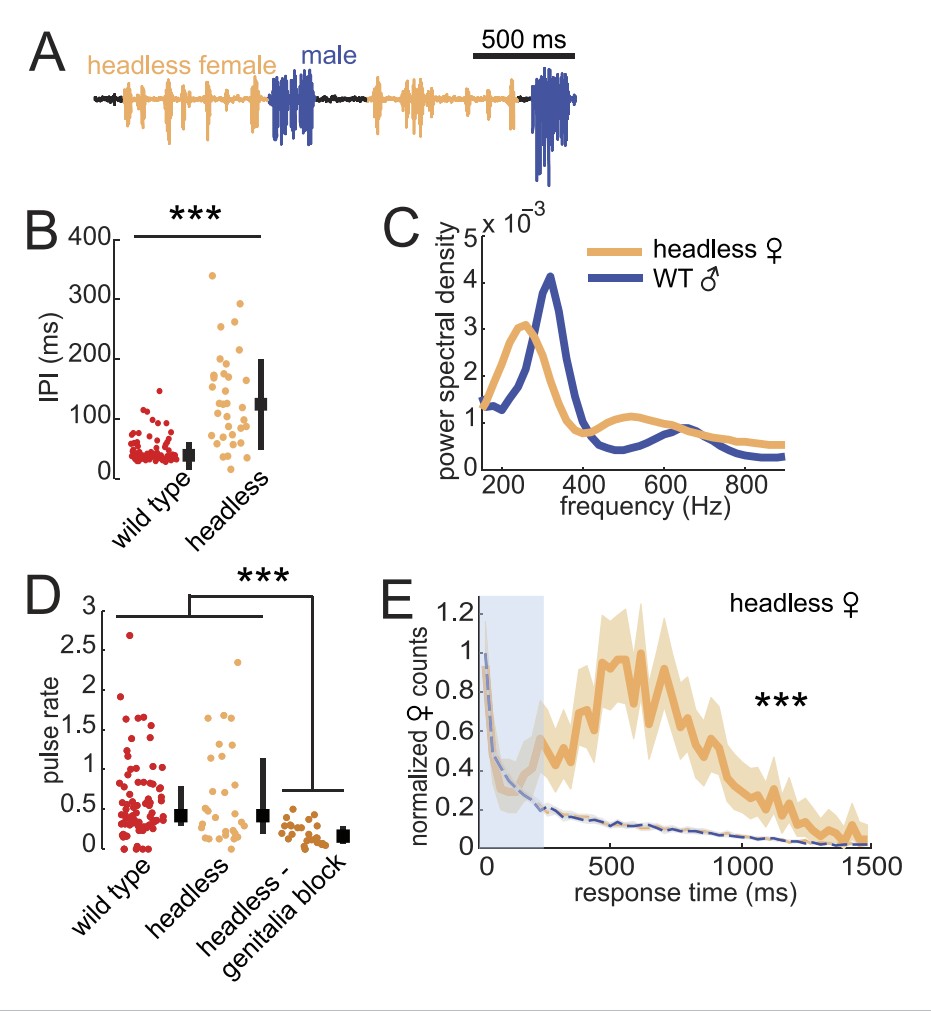

**Figure 6**. Testing the sufficiency of abdominal sensory inputs for female song production and timing. (**A**) Headless female song (orange) and WT male song (blue). (**B**) Median IPI of headless female song (orange, n = 33) compared with IPI from WT females (red, n = 83). Black squares indicate population mean and black bars s.d. (Student's *t*-test, ***p < 0.001). (**C**) Power spectral density of pulses from headless females (orange, n = 21,481 pulses) or their WT male courtship partner (blue, n = 11,136 pulses). (**D**) Median pulse rate for headless females (orange, n = 33) and headless and genitalia blocked females (dark orange, n = 31), compared to WT females (red, n = 83). Black squares indicate population median and black bars IQR (Kruskall–Wallis test, ***p < 0.001). (**E**) Normalized female response time distribution for headless females (orange, n = 1430, n = 33 courtships), compared with distributions from shuffled male bout (blue dashed) and female pulse (orange dashed) intervals (***p < 0.001 K-S test). Shading around each response time distribution represents bootstrapped 95% confidence intervals.

during courtship. We then compared female response time distributions for AC females with block to either abdominal segments A3–A5 or the genitalia (*Figure 7H*). We found a larger shift in the response time distribution when blocking the genitalia (KL divergence between AC distribution and AC genitalia block distribution = 0.36) vs when blocking the upper abdomen (KL divergence between AC distribution and AC upper abdomen block distribution = 0.12). We, therefore, propose that while both forms of male contact are required for normal song rates, female song timing relies more strongly on the timing of male proboscis contact with her genitalia. This represents a novel mechanism for coordinating duets. We hypothesize that male contact cues (both tarsal vibration and proboscis licking) are detected via mechanosensory neurons, as the female abdomen and genitalia are covered in mechanosensory machrochete bristles (*Fabre et al., 2008*). In contrast, gustatory receptors are expressed in multidendritic neurons that tile the abdominal body wall

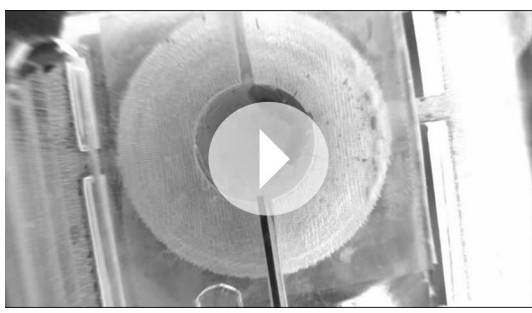

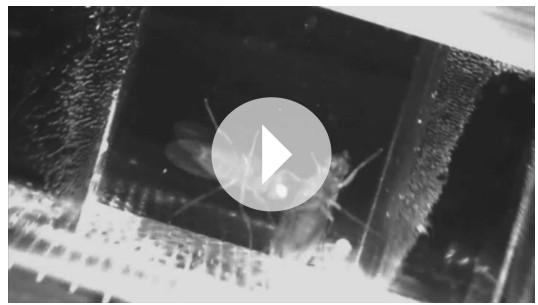

**Video 2.** *D. virilis* headless female duetting behavior. Representative example of a naive male paired with a naive decapitated female in one channel/chamber of the multi-channel song recording system. Video is acquired at 15 Hz.

**Video 3.** High-speed video of *D. virilis* duetting behavior. Representative example of courtship between a naive *D. virilis* male and female pair in a 1 × 1 × 0.5 cm clear plastic chamber, with microphone placed adjacent to the chamber. Video is acquired at 60 Hz.

(*Park and Kwon, 2011*), but none of these neurons detect external chemical cues. Our videos, however, did not reveal which subsets of bristles are likely to be responsible for detecting the male contact cues.

## *D. virilis* female song production circuits are not sexually differentiated

In *D. melanogaster*, song production circuits are sexually differentiated or dimorphic (*von Philipsborn et al., 2011*); only males of this species produce song. Females are typically silent but can be forced to produce (aberrant) song via artificial activation of *fruitless*-expressing neurons (*Clyne and Miesenbock, 2008*). This sexual dimorphism relies on male-specific isoforms of both the *fruitless* and *doublesex* genes (*Demir and Dickson, 2005*; *Manoli et al., 2005*; *Rideout et al., 2010*), the regulation of which are conserved in *virilis* (*Yamamoto et al., 2004*; *Usui-Aoki et al., 2005*). We, therefore, expected that the disparate male and female song behaviors in *D. virilis* should also be sexually dimorphic. To test this hypothesis, we paired either two females or two males in behavioral chambers. We did not observe any song production in pairings between two females (data not shown). However, similar to studies in *D. melanogaster* (*Villella et al., 1997*; *Pan and Baker, 2014*), we found that males will court another male. We observed two types of song in these interactions (*Figure 8A*): the courting male produced male-typical songs (blue), while the male being courted produced a secondary song that appeared more female-like (green), with longer IPIs (*Figure 8B*, compare to *Figure 1E*). However, the fundamental frequency of the pulses of the courted male's song remained male-like (*Figure 8C*). This secondary song (*Suvanto et al., 1994*), similar to female song, was generated by bilateral wing vibration (*Video 4*). Strikingly, we found response times of the courted male (secondary) song relative to the courting male (primary) song to match the distribution of female response times (compare *Figure 8D* with *Figure 2A*). This implies that the male nervous system contains separable circuits for song production, which are each activated in a context-dependent fashion. When the male is courting a target, he produces 'male' or primary song bouts, but when he is being courted, he produces 'female' or secondary song at the appropriate (female-like) response delay (see *Video 5* e.g., of males alternating between courting and being courted).

To determine how the response timing of male secondary song is regulated, we again combined higher speed video and acoustic recordings. Similar to male–female courtship, tarsal vibration and proboscis licking are most predictive of the occurrence of secondary song (*Figure 8F*), with the GLM filters peaking immediately prior to secondary song (*Figure 8G*). Likewise, these behaviors are also predictive of male primary song (*Figure 8H*), and the shapes of filters were similar for male–male vs male–female interactions (compare *Figure 8I* with *Figure 7F*). Predictability was reduced when compared with models for predicting song in male–female interactions (e.g., compare *Figure 7C* with *Figure 8F*) and was not substantially enhanced with a two-variable model (data not shown). We conclude that male secondary song timing is best correlated with, similar to female song timing, both male tarsal and proboscis contact with the abdomen and genitalia. Therefore, we propose that sexually monomorphic (or undifferentiated) sensory bristles and neural circuits drive female duetting behavior in *D. virilis*.

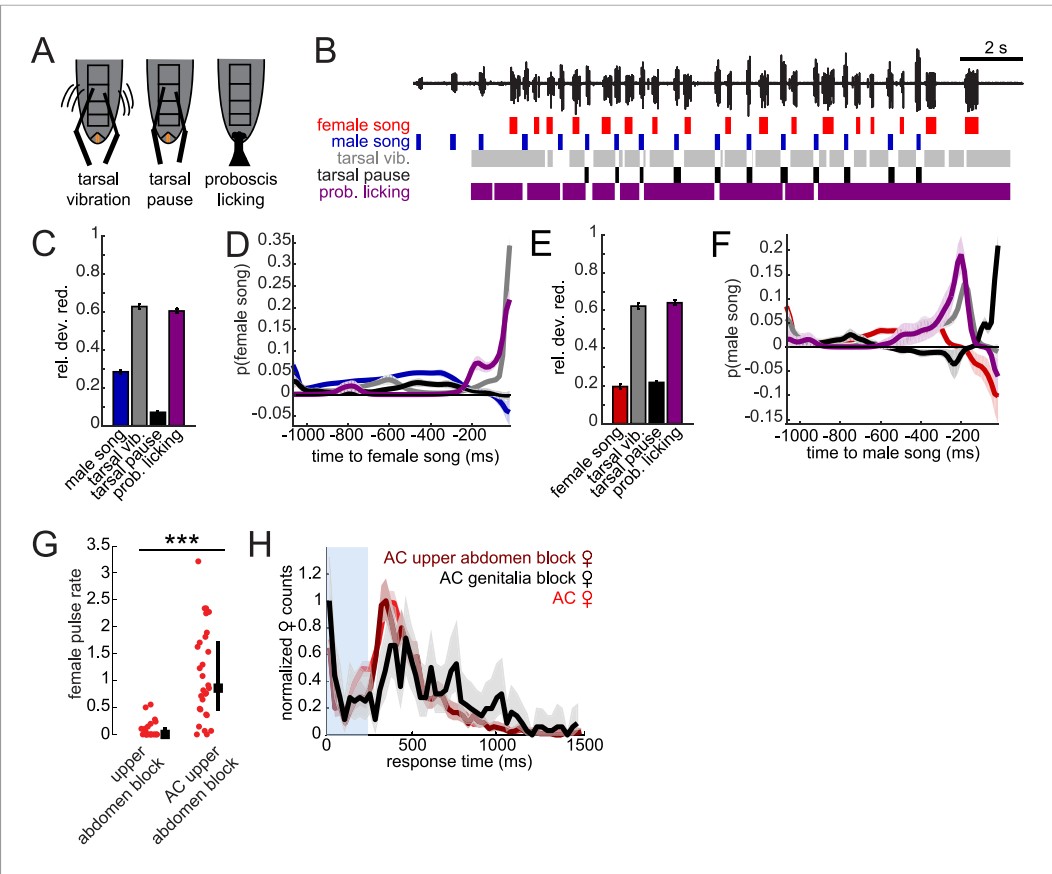

**Figure 7**. Male contact with the female abdomen and genitalia predicts song timing. (**A**) Schematic of annotated male contact behaviors (see also *Video 3*). (**B**) Example of a duet between a WT male (blue) and female (red), accompanied by annotation of male tarsal contact with the abdomen (tarsal vibration (gray) and tarsal pause (black); tarsal pause denotes male tarsal contact with the female abdomen, but without vibration) and proboscis licking of the genitalia (purple). GLM analysis was used to predict the presence/absence of male or female song based on the time course of annotated behaviors during courtship interactions. Model performance (indicated by relative deviance reduction) reveals the behaviors most predictive of female song (**C**) and male song (**E**) (see 'Materials and methods'). GLM filters reveal the times at which each behavior is most predictive of female song (**D**) or male song (**F**, n = 7 courtships with approximately 200 instances of both male and female song). Error bars and shading represent bootstrapped s.e.m. estimates. (**G**) Females with their upper abdomens (tergites A3–A5) blocked produce very little song (n = 23 courtships), compared with females whose aristae were additionally cut (n = 29 courtships, ***p < 0.001 Wilcoxon rank-sum test). Black squares represent median and black bars the IQR. (**H**) Normalized female response time distributions for AC females (red, reproduced from *Figure 1H*), AC-upper abdomen blocked females (dark red, n = 1892, n = 29 courtships, Kullback–Leibler Divergence = 0.12 between AC and AC-upper abdomen block curves) and AC-genitalia blocked females (black, n = 483, n = 18 courtships, K-L Divergence = 0.36 between AC and AC genitalia block curves); all females were paired with WT males. Shading around each response time distribution represents bootstrapped 95% confidence intervals.

The following figure supplement is available for figure 7:

**Figure supplement 1**. Two-variable GLM for male–female pairings.

## Discussion

In this study, we combine targeted sensory manipulations, high-throughput song recording and analysis, and statistical modeling to provide the first quantitative characterization of acoustic duetting in a Drosophilid species. This is significant because, as a Drosophilid, *virilis* shares developmental similarities with *melanogaster* (*Kuntz and Eisen, 2014*) and in addition has had its genome fully sequenced and annotated (*Drosophila 12 Genomes Consortium et al., 2007*). This makes feasible the development of

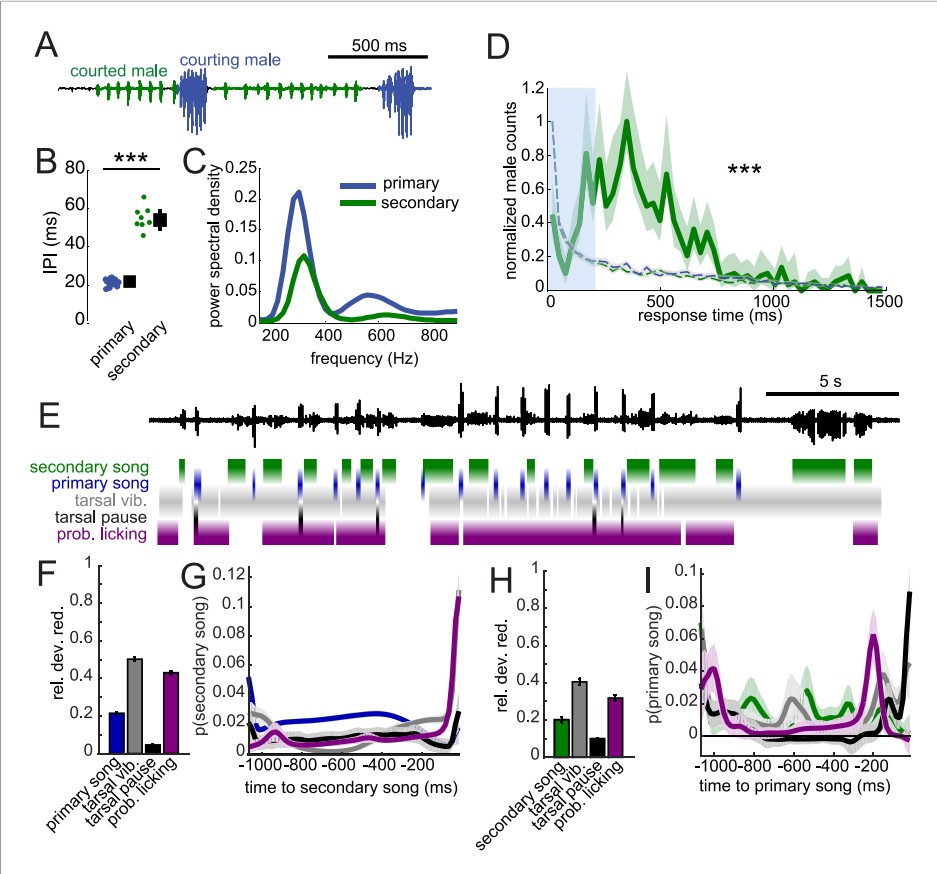

**Figure 8**. When courted by another male, males can produce female-like duets. (**A**) Song produced in pairings between two WT males. Combining video and acoustic recording (*Video 4*) reveals that male primary song is produced by the courting male (blue), and male secondary song is produced by the courted male (green). (**B**) IPI of male secondary song (green, n = 8) compared with the IPI of male primary song (blue, n = 13, Wilcoxon rank-sum Test, ***p < 0.001). (**C**) Power spectral density of pulses from male secondary song (n = 6349, green) and male primary song (n = 1309 bouts, dark blue). (**D**) Normalized distribution of courted male response times to male primary song (green, n = 766, n = 53 courtships) compared to response times from shuffled data (K-S Test, ***p < 0.001). (**E**) Example of a male–male duet, accompanied with video annotation of tarsal and proboscis movements of the courting male. GLMs were used to predict the presence/absence of male secondary song and male primary song from the temporal pattern of annotated behaviors preceding each time point during courtship interactions. GLM performance indicates the male behaviors most predictive about male secondary song (**F**) and male primary song (**H**). GLM filters reveal the times at which each behavior is most predictive of male secondary song (**G**) or male primary song (**I**). (n = 7 courtships with 107 and 127 instances of male secondary and primary song, respectively). Error bars and shading in F-I represent bootstrap estimates of the s.e.m.

genetic (*Bassett and Liu, 2014*) and neural circuit tools (*Simpson, 2009*) to resolve the mechanisms underlying duetting. Such tools will allow us to determine, for example, if genes such as *fruitless* and *doublesex*, known to be important for the establishment of sexually dimorphic and courtship-related behaviors in *melanogaster* (*Kimura et al., 2008*), play different roles in a species in which both males and females are capable of song production. Our study also uncovered the sensory cues and putative neural mechanisms that orchestrate duetting in *virilis*. We can address these mechanisms by, for example, targeting neurons (via genetic methods) in *virilis* that are homologous to recently mapped song pathway neurons in *melanogaster* (*von Philipsborn et al., 2011*). These experiments should reveal for the first time how the function and modulation of the song motor pathway differs between males and females in a duetting species. Because *virilis* is separated from *melanogaster* by >40 million years of evolutionary time (*Drosophila 12 Genomes Consortium et al., 2007*), the development of these tools should additionally provide insight into the evolution of new behaviors such as female song production.

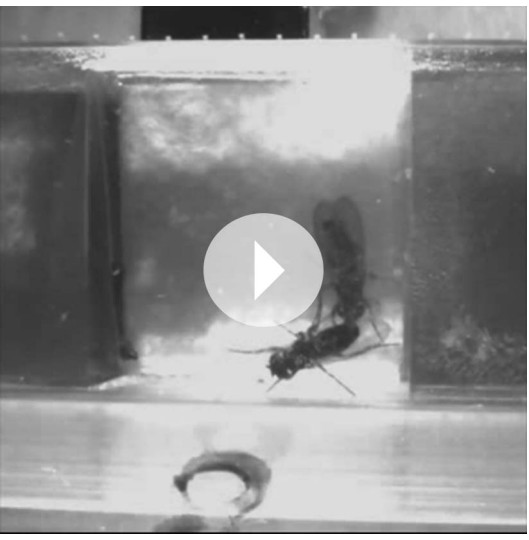

**Video 4.** High-speed video of male–male behavior. Representative example of an interaction between two naive *D. virilis* males in a 1 × 1 × 0.5 cm clear plastic chamber, with microphone placed adjacent to the chamber. Video is acquired at 60 Hz.

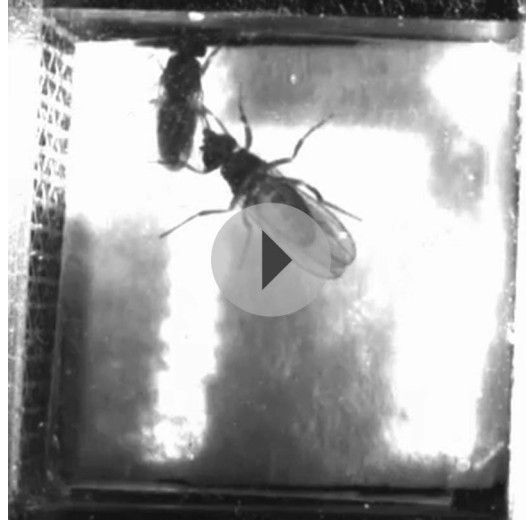

**Video 5.** High-speed video of male–male behavior. Another example of an interaction between two naive *D. virilis* males, showing role-switching behavior between courting and being courted.

We also establish with this study that duetting in *D. virilis* relies on multisensory cues. Our experiments using song playback (to test the sufficiency of auditory cues) and headless females (to test the sufficiency of tactile cues) showed that neither auditory nor tactile cues alone fully recapitulated female song responses (e.g., response time distributions did not wholly match those of intact females interacting with males). In addition, we showed that interfering with song production (e.g., blocking male tactile inputs to the female abdomen and genitalia) could be overcome by additional removal of the auditory receiver (arista). These results taken together suggest that auditory and tactile information are integrated to drive song production and coordination in females. This may be similar to the integration of multimodal signals observed in frogs during territorial behaviors (*Narins et al., 2005*). Experiments that target the *virilis* song pathway for neural recordings will reveal if this pathway is directly sensitive to both auditory and tactile information (both types of stimuli can be easily applied in a preparation fixed for in vivo recordings [*Murthy and Turner, 2010*]). These experiments should also reveal if auditory information is combined linearly (simple integration) vs nonlinearly with tactile cues.

Other studies in insects have found a role for tactile cues in some social (but non-courtship) behaviors (*Rogers et al., 2003*; *Ramdya et al., 2014*); however, ours is the first demonstration of a role for tactile cues in acoustic duetting in any animal species. Moreover, our GLM analysis revealed that it is the precise timing of the male's contact with the female abdomen and genitalia that is the biggest predictor of her song timing. These experiments and analysis not only expose a completely novel mechanism for coordinating acoustic behavior between individuals but raises the question of what advantage incorporating tactile information provides to the female. Because removing any individual sensory cue to the *virilis* female (e.g., tactile cues sensed either via the genitalia or the upper abdomen) disrupts female song production, it seems unlikely that a male provides multiple cues in the event that one of his signals fails—rather, females may select for males that can reliably provide multiple timing cues. This idea is consistent with studies showing that generating multiple cues is energetically costly for the sender (*Partan and Marler, 2005*). Moreover, studies of decision-making in higher systems show that different sensory channels provide independent sources of information for directing behavior (*Fetsch et al., 2012*). If this is true also for *virilis*, females may be better able to perform the duet by relying on multiple channels.

The only sensory manipulation performed in this study that disrupted duetting while leaving song production rates intact was deafening the male (*Figure 2*). However, this manipulation did not affect time to copulation, which suggests that neither do males need to hear the female's song nor do females require a properly timed duet for mating. On the other hand, our experiments on virgin vs mated females suggest that female song production is a positive cue, because it is associated with her receptivity state. It is possible that our experimental conditions (pairing one male with one female) did not uncover the role duetting normally plays in mating. Studies in other model systems suggest the female song production evolved in species that experience high levels of reproductive competition (*Clutton-Brock, 2007*). Examining interactions between either two males and one female or two females and one male, for example, could reveal the relationship between duetting and mate selection in *virilis*; such experiments are feasible using *Drosophila* behavioral chambers with multiple microphones to separate the signals of multiple singers (*Coen et al., 2014*). Finally, because humans can integrate both auditory and tactile cues during speech perception (*Gick and Derrick, 2009*), we propose that, more broadly, tactile cues may be critical for acoustic communication across the animal kingdom. Mapping the *virilis* circuits that detect tactile information, and then relay this information to the song production pathway, should therefore provide general insights into how precisely timed behaviors, such as song and speech production, are regulated.

Finally, our data suggest that neural circuits for female duetting (from circuits that detect tactile cues all the way to circuits that direct the production of song pulses) are not sexually dimorphic or differentiated between males and females; this result sharply contrasts with the observed sexual dimorphism of song production circuits in *D. melanogaster* (*Clyne and Miesenbock, 2008*; *von Philipsborn et al., 2011*). However, we did not find conditions under which females produce male-like songs, suggesting that *virilis* male primary song production circuits are sexually differentiated. This discovery should aid in identifying the circuits for duetting in *virilis* (in other words, some song circuit elements should be present in both males and females, while others should only be present in males). Moreover, during male–male courtship, *virilis* males can switch between the production of female-like secondary or male-like primary song based on their role as courtee or courter, respectively (*Video 5*). While some insects participate in sexual mimicry (imitating the song of the partner) during male–female courtship (*Luo and Wei, 2015*), ours is, to the best of our knowledge, the first demonstration of context-dependence of duetting behavior in any animal species. Therefore, dissecting the neural circuit basis for song production in *D. virilis* promises to reveal novel insights into how animals rapidly modulate behavior in response to changing sensory feedback.

## Materials and methods

### Fly stocks and husbandry

We tested *D. virilis* strains 15010-1051.47, 51, 09, 48, 49, and 52 (UCSD Stock Center, San Diego, CA) in our song-recording chambers. We chose strain 15010-1051.47, derived from Hangchow, China, for all experiments in this study, because both males and females of this strain produced the most song during courtship. Flies were maintained on standard medium at 20°C, on a 16-hr light:8-hr dark cycle (*Suvanto et al., 1999*). For behavioral assays, virgin males were housed individually, while virgin females were group housed. All flies were aged 10–20 days; this is the time required to reach sexual maturity (*Isoherranen et al., 1999*). Behavioral assays were performed between 0–4 hr ZT and at approximately 22°C.

### Sensory manipulations

Aristae and antennae were removed by plucking with tweezers (Fine Science Tools, Foster City, CA). Distal fore-tarsi were removed by cutting with scissors (Fine Science Tools, Foster City, CA). To block contact with the female genitalia, we covered the genitalia and a portion of the A6 tergite with a low-melting temperature paraffin wax. We confirmed that results were similar whether we used wax or a UV-curable glue. To block contact with the female upper abdomen, we covered only tergites A3–A5. Visual cues were removed by turning off the lights in a windowless room and covering the recording system in blackout curtains. In this condition, red LED lighting was used for observations. We performed all physical manipulations on $CO_2$-anesthetized flies a minimum of 12 hr prior to behavioral assays. The only exception was for decapitation, which was performed on females at least 30 min prior to recording.

## Song recording

Courtship songs were recorded at 10 kHz on a 32-channel apparatus (*Arthur et al., 2013*). Mature flies were mouth aspirated into plastic chambers modified from the design in (*Arthur et al., 2013*). Modified chambers were larger (2-cm diameter) and raised, to prevent direct contact with the microphones. These changes were made to accommodate *virilis* flies, which are larger and sing louder than *melanogaster* flies. To encourage two males to interact with each other, a small piece of paper was added to the chamber to reduce the available space. Acoustic behaviors were recorded for 20 min, and it was noted if copulation occurred during this period.

## Song segmentation

We wrote annotation software (FlySongSegmenter-Virilis [FSS-V]) that performs a continuous wavelet transformation (using the frequency B-spline wavelet) on the raw song trace to extract the spectral features (between 100 and 900 Hz) of the song as a function of time. See *Figure 1—figure supplement 3*. For each time point in the wavelet transform, we computed the likelihood that it was generated from one of four previously assembled template distributions. These template distributions were created from manually annotated data for male, female, and overlapping pulses. The wavelet transforms corresponding to the pulse locations for each of these groups were then collated and rotated into a new orthogonal coordinate frame via principal components analysis. The distributions of projections along the modes were fit to a Gaussian mixture model through an expectation maximization algorithm. Thus, each template consists of a mean wavelet spectrum, an orthogonal basis set, and a set of distributions along each of these newly created axes. An identical procedure was performed to create a noise (N) template as well. However, for this case, we created a new noise template from the low-amplitude fluctuations in each data set separately, allowing us to account for differing noise conditions. Given these templates, we defined the likelihood that a wavelet was drawn from one of these templates, T, by transforming it into the appropriate template coordinate system and calculating $p(x_i|T)$ for each coordinate value, $x_i$. The log-likelihood is simply the sum of the logarithms of these values. The end result of this process was to have a set of 4 log-likelihoods as a function of time (male [M, blue], female [F, red], overlap [O, green], and noise [N, black]). If we assume a uniform prior for the template distributions, then, $p(T|x(t))$ is simply proportional to the likelihood value, so we can then assign a probability to each of the templates as a function of time. We first eliminated all noise from the time series, zeroing out all segments where $p(N|x(t)) > 1/2$. We then looked at each connected segment of non-noise data separately. If $p(M|x(t)) + P(O|x(t)) > p(F|x(t))$ and the segment were at least 100 ms in length, we assigned that segment to M. All remaining segments at least 15 ms in length were assigned to F. Lastly, we looked within each of the M regions for overlapping pulses, where both the male and the female were singing. If $p(F|x(t)) + P(O|x(t)) > 0.8$ over a region at least 15 ms in length, we labeled that as a male region with a female pulse. The automated code is available on github (https://github.com/murthylab/virilisSongSegmenter). Lastly, these pulse calls were scanned by visual inspection, correcting mislabeled pulses by hand. Song segmented by this method was compared to manually segmented song from a WT data set that was not previously used to inform the likelihood models. Sensitivity, positive predictive value, and the harmonic mean (F) were calculated as previously described (*Arthur et al., 2013*).

## Video recording and annotation

To correlate song production with a particular singer (e.g., male vs female), we used a USB2.0 CMOS 1280 × 1024 Monochrome Camera (Thor Labs, New Jersey, USA) with 5-mm EFL lens, and uc480Viewer recording software was used to record single-courting pairs at ~15 Hz. Audio and video data were synced using iMovie. To record fly behavior (in particular, male contact with the female abdomen) during duetting, we used a Point Grey Monochrome Camera with a 5-mm EFL lens to record single-courting pairs at 60 Hz in a 1 cm × 1 cm × 0.5 cm clear plastic chamber. Audio and video data were synced using iMovie. These videos were annotated for onset and offset of three behavioral features (tarsal contact with vibration, tarsal contact without vibration (pausing), and proboscis contact) using ANVIL (www.anvil-software.org); we annotated only video segments during which both males and females produced song. Male and female songs were annotated in MatLab (Mathworks Inc).

## Playback experiments

Synthetic male courtship song (a recorded male bout was smoothed using a 300-Hz Butterworth low-pass filter; a single stimulus contained this bout repeated at 6× at 1.2-s intervals; stimuli were delivered every 30 s for 10 min) was delivered to females, paired with WC males, in a modified courtship chamber (a 2-cm diameter hole was cut into the top of the plastic chamber and replaced with mesh). Song was delivered via Koss earbud speakers, and earbuds were mounted above each chamber and oriented at a 45° angle toward the arena. Song intensity was calibrated to match song recorded in the same chambers (between an intact male and female pair). We were unable to score most female song that occurred at the same time as the playback stimulus (due to differences in intensity between the playback and the female song). Therefore, we also calculated female response time from the offset of the male artificial bout, which ignores potential overlaps (see inset in *Figure 5B,C*). This is the only instance in all analyses in this study where overlaps were ignored.

## Latency to mating assay

Male/female pairs were aspirated into clear plastic chambers, each $1 \times 1 \times 0.5$ cm. Pairs were manually observed for 30 min, and copulations were recorded every minute. Manipulated flies (e.g., AC) and their corresponding controls (e.g., flies with intact aristae, but held on the $CO_2$ pad for the same amount of time) were observed simultaneously. Experiments that required observation in the dark were performed in a dark room under red LED lighting. Only pairs for which there was visible male wing vibration were scored. To generate mated females, females were paired with males until copulation occurred, at least 24 hr prior to pairing with virgin males.

## Statistical analyses

All statistical analysis was performed in Matlab (Mathworks, Inc.).

## Song statistics

A male bout must contain at least four concurrent male pulses, with IPIs of less than 25 ms. Female bouts consist of successive pulses with IPIs <100 ms. IPI values were calculated as the time between pulses with a threshold of 100 ms for males and 500 ms for females and reported as a median per individual. Male bout and female pulse rates were calculated as the number of bouts or pulses divided by total courtship time in seconds (Hz), where total courtship time is the time between the first pulse in the recording (male or female) and the last pulse (male or female). Because there can be long stretches of silence during a recording, song rates report how much singing occurs overall within a recording, whereas IPI reports the rate of pulsing when singing occurs. We chose to quantify female pulse rates (as opposed to bout rates) due to the highly variable structure of female bouts. Recordings with less than 20 s of either male or female song were assigned a male bout rate or female pulse rate of 0 Hz. The square roots of female pulse rates are plotted in all figures (to temper outliers), but all statistics were performed on raw data. A GLM (for details see next section) was used to determine the significance of song rate differences between virgin and mated females, independent of courtship success. Instances of overlaps between male and female song are reported as a percent (number of overlaps/number of male bouts) for each courtship pairing. For response times, we calculated the delay from the onset of a male bout to the previous female pulse within 1.5 s (male response time) or the delay from the onset of a male bout to the following female pulse within 1.5 s (female response time). This analysis includes regions of male and female overlap. We also randomly shuffled either female IPIs or male IBIs and then calculated male and female response times. Confidence intervals were generated with 500 bootstrapped permutations. The two-sample Kolmogorov–Smirnov test was used to determine if the difference between response time curves was statistically significant. We report median values and interquartile range (IQR) for non-normally distributed data and mean values and standard deviation for normally distributed data. Male secondary song elicited during male–male interactions was analyzed in the same manner as female song above.

## Generalized linear model analysis for sensory manipulations and mating state

To determine the sensory pathways influencing song production rate in males and females, we fit a GLM to the behavioral data. Since pulse/bout rates follow Poisson statistics (rates are bounded

between 0 and infinity), we used an exponential link function in the model. The presence/absence of a sensory channel was coded as 1/−1. To ensure that the GLM coefficients reflect the relative impact of removing a sensory channel, we first z-scored the data. Fitting was performed using Matlab's *fitglm* function, and statistics and errorbars were extracted from the output of that function. The same approach was used to demonstrate differences in song rates dependent on the female mating state. For each pair, the state of the female (virgin/mated) and the success of the courtship (no copulation/copulation) were coded in a two-variable matrix (represented as −1/1, respectively). Statistical output of the fitglm function (from MatLab) is reported.

### Generalized linear model analysis for annotated videos

To determine the behaviors that control the timing of song, we fit GLMs, as in *Coen et al. (2014)*, to predict the presence/absence of male song or female song from annotated behaviors during a courtship interaction. To learn about *when* a behavior is predictive about song, we used the temporal pattern of behaviors for prediction. For each time point in male or female song, we used the temporal pattern of behaviors in a window of ~1 s (64 frames at 60 fps) preceding that time point for predicting song. This yielded GLM filters—a sequence of weights whose magnitude indicates the importance of each time point in the behavioral history. Since the predicted variable is binary (song/no song), we used a logistic link function for the GLM. The annotated behaviors are strongly correlated both in time (autocorrelation) as well as with each other (cross-correlation). We, therefore, used a GLM with a sparse prior that penalized non-predictive weights, which would have large magnitudes merely due to correlations in the data (*Mineault et al., 2009*; *Coen et al., 2014*). Model performance (relative deviance reduction) was evaluated using cross-validation; that is, model parameters were fitted from 80% of the data, and the model was tested with the remaining 20% of the data (random subsampling). To obtain a bootstrap estimate of the standard error of the mean for the GLM filters and GLM performance, we re-ran this cross-validation procedure 1000 times with a randomly selected 75% of all data. We started by fitting single-variable models (e.g., using only tarsal contact as a predictor) and determined the most predictive features. To rule out that including more parameters substantially improved performance, we took the best predictor and added any of the remaining three features to the model. However, this never increased performance by more than 14%. Code for sparse GLM analysis is available on GitHub (https://github.com/murthylab/GLMvirilis).

## Acknowledgements

We thank B Arthur and D Stern for assistance in establishing the song recording system, G Guan for technical assistance, and T Perez for assistance with latency to mating experiments. We thank Asif Ghazanfar, David Stern, Adam Calhoun, Dudi Deutsch, and the entire Murthy lab for thoughtful feedback and comments on the manuscript. JC is supported by the DAAD (German Academic Exchange Service), GB is supported by NIH Grant GM098090, and MM is funded by the Alfred P Sloan Foundation, the Human Frontiers Science Program, the McKnight Endowment Fund, the Klingenstein Foundation, an NSF CAREER award, an NIH New Innovator Award, and an NSF EAGER BRAIN Initiative award.

## Additional information

### Funding

| Funder | Grant reference | Author |
| --- | --- | --- |
| Human Frontier Science Program | RGY0070/2011 | Mala Murthy |
| National Science Foundation (NSF) | CAREER IOS-1054578-005 | Mala Murthy |
| German Academic Exchange Service | postdoctoral fellowship | Jan Clemens |
| National Institutes of Health (NIH) | GM098090 | Gordon J Berman |
| Alfred P Sloan Foundation | | Mala Murthy |
| McKnight Endowment Fund for Neuroscience | | Mala Murthy |

| Funder | Grant reference | Author |
|---|---|---|
| Esther A and Joseph Klingenstein Fund | Klingenstein-Simons fellowship | Mala Murthy |
| Simons Foundation | Klingenstein-Simons fellowship | Mala Murthy |
| National Institutes of Health (NIH) | New Innovator Award | Mala Murthy |
| National Science Foundation (NSF) | CAREER award | Mala Murthy |
| National Science Foundation (NSF) | EAGER BRAIN Initiative award | Mala Murthy |

The funders had no role in study design, data collection and interpretation, or the decision to submit the work for publication.

## Author contributions

KML, Conception and design, Acquisition of data, Analysis and interpretation of data, Drafting or revising the article; JC, GJB, Analysis and interpretation of data, Drafting or revising the article; MM, Conception and design, Analysis and interpretation of data, Drafting or revising the article

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
