## [Decision Letter]

Thank you for sending your work entitled “Acoustic duetting in *Drosophila virilis* relies on the integration of auditory and tactile signals” for consideration at *eLife*. Your article has been favorably evaluated by a Senior editor and three reviewers, one of whom is a member of our Board of Reviewing Editors.

The following individuals responsible for the peer review of your submission have agreed to reveal their identity: Ronald L Calabrese (Reviewing editor), Julie Simpson (peer reviewer), and Darcy Kelley (peer reviewer).

The Reviewing editor and the other reviewers discussed their comments before we reached this decision, and the Reviewing editor has assembled the following comments to help you prepare a revised submission.

The authors present a very intriguing and thorough behavioral analysis of acoustic communication in adult *Drosophila virilis* that shows that acoustic communication is two way; males and females duet. These duets are a part of courtship that leads to successful matings. They are synched in the female mainly by tactile cues which the male provides rather than by auditory cues, though auditory cues do play a role. In the males auditory cues play a bigger role in synchronization but tactile cues are also important. These conclusions are based on extensive behavior tests with sensory modifications and the application General Linear Models (GLMs). Most intriguing is that males can also produce “female-like” songs, elicited by tapping, when isolated with a courting male. Thus, both sexes possess neural circuits capable of producing this type of courtship song. The writing in general is clear and concise. The figures are a bit busy (there is no need for this as there are no figure limitations) but clear in general. The modeling is well conceived and based on established procedures and it appears competently and prudently applied. The Methods section is appropriate but could provide access to the code used in the modeling and analysis. These results have important implications for analysis of acoustic communication in animals by pointing out a role for multisensory cues in the acoustic signaling, and they could serve as an entry point for further mechanistic studies in an important model system.

The consensus among the reviewers is that the paper is suitable for publication in *eLife* with no additional experiments but significant clarification of text on issues raised below and addition of an introductory figure.

1) Duets: The authors show that singing is synchronized between male and female and the synchronization is achieved multimodally by female responses to male input mostly tactile. In this light the male triggers a bout (train) of quite variable female song that may overlap with his next song bout. A general reader may have a different concept of a duet as mutual acoustically driven performance with each singer producing a precise song component as seen for example in human singers. The authors need to do a better job in the Introduction and Discussion explaining the context of their work and how the mutual singing they describe conforms to the concept of duets in the ethological literature. Then the multimodal nature of the synchronizing mechanism can be more plainly seen as novel.

It would be helpful to start off the Results section with an introductory figure (there are no figure limitations in *eLife*) showing a picture of courting male and female flies together with complete series as in Figure 5, but with parts expanded a bit like Figure 1, of duets (from a few different pairs) produced during the courtship interaction. The authors can use this introductory figure to describe the features of the signals that they use to define pulses and inter-pulse intervals, bouts etc. This figure should include a sequence where male and female song overlap if indeed they do under normal circumstances.

2) Overlap of Male and Female Song: There was some confusion about the issue of song overlap and how this was handled in the data analysis that the authors should clarify. For example in the playback experiments of Figure 4 was the considerable overlap excluded (“Moreover, due to issues (in these experiments) with identifying female pulses that overlap with the male playback (see Methods), we also calculated female response times from the end of the male bout (Figure 4 inset).”)? How often did the songs overlap in the courting data set? In the courtship song sequence of Figure 5 there appears to be no overlap at all. Is this correct and truly typical in the data set? If overlap occurs are the songs disentangled or is the data excluded. If overlap is excluded from the mutual singing experiments would not alternation be enforced by the analysis especially given the variable latency between male song and female ‘response’? Reference to any potential overlap should be made in the introductory figure and rationale for handling overlap in the analysis expanded.

3) Sexually Monomorphic: The sexually dimorphic or monomorphic terminology may be difficult for the general reader to follow. The authors need to put this terminology more firmly in the context of standards in the literature and what is known about duetting in other species or consider alternatives. A particular type of song may be elicited by tapping on the abdomen, and this song can be produced by females or males. Thus the authors might consider using female-typical for this sort of song and sexually differentiated to deal with the issue that females don't produce male-type song patterns.

4) Other Concerns:

A) Male singing seems to be important for courtship success. Tapping may also important for courtship success, but is duetting itself? Does the total amount of the song matter or the timing? Is there a way to leave tapping intact but de-synchronize the song?

B) Male performance is altered by deafening (Figure 1), but it isn't clear if he needs to hear his own song or the female's.

C) The authors might want to cite Peter Narins' studies on visual/auditory integration in tungara frogs as another example of multimodal integration. Peter described the phenomenon as “binding” an idea worth considering in terms of the value of multiple sensory inputs.

D) The authors seem to assume that duetting is derived rather than ancestral. For those readers not au courant with Drosophilid phylogeny, it would be helpful to explain the basis of this assumption. Presumably it reflects the prevalence of song in males alone but is enough known? The observation that males sing and females listen can obscure the more general finding that sexual dimorphisms are lost more often than they are gained (65).

E) The authors state that competition is a possible driving force for the emergence of female songs and propose experiments using two males and a female. Might not the competition be between females?

F) The authors indicate that one of the promises of this system is to dissect the genes and neural circuitry in *virilis* (using clues from melanogaster) that control multimodal integration involved in duetting. The authors might expand and be specific about why/how this system will allow forward progress on this interesting topic.

---

## [Author Response]

*The authors present a very intriguing and thorough behavioral analysis of acoustic communication in adult* Drosophila virilis *that shows that acoustic communication is two way; males and females duet. These duets are a part of courtship that leads to successful matings. They are synched in the female mainly by tactile cues which the male provides rather than by auditory cues, though auditory cues do play a role. In the males auditory cues play a bigger role in synchronization but tactile cues are also important. These conclusions are based on extensive behavior tests with sensory modifications and the application General Linear Models (GLMs). Most intriguing is that males can also produce “female-like” songs, elicited by tapping, when isolated with a courting male. Thus, both sexes possess neural circuits capable of producing this type of courtship song. The writing in general is clear and concise. The figures are a bit busy (there is no need for this as there are no figure limitations) but clear in general. The modeling is well conceived and based on established procedures and it appears competently and prudently applied. The Methods section is appropriate but could provide access to the code used in the modeling and analysis*.

We have split up some of the figures now to make them less busy. We have also now put our code for *virilis* song segmentation and for the sparse GLM on Github (https://github.com/murthylab/virilisSongSegmenter and https://github.com/murthylab/GLMvirilis), and we provide the link to the website in the Methods section.

*These results have important implications for analysis of acoustic communication in animals by pointing out a role for multisensory cues in the acoustic signaling, and they could serve as an entry point for further mechanistic studies in an important model system*.

*The consensus among the reviewers is that the paper is suitable for publication in* eLife *with no additional experiments but significant clarification of text on issues raised below and addition of an introductory figure*.

*1) Duets: The authors show that singing is synchronized between male and female and the synchronization is achieved multimodally by female responses to male input mostly tactile. In this light the male triggers a bout (train) of quite variable female song that may overlap with his next song bout. A general reader may have a different concept of a duet as mutual acoustically driven performance with each singer producing a precise song component as seen for example in human singers. The authors need to do a better job in the Introduction and Discussion explaining the context of their work and how the mutual singing they describe conforms to the concept of duets in the ethological literature. Then the multimodal nature of the synchronizing mechanism can be more plainly seen as novel*.

Thank you for this suggestion. We have added text and additional references to the Introduction describing the criteria for considering an acoustic interaction to be a duet and describing that each individual’s component of the duet need not be stereotyped. We have also expanded the first two paragraphs of the Results section to better describe the acoustic interaction in *virilis.* We agree that this additional text now strengthens the argument that the use of multisensory cues should facilitate the temporal coordination involved in duetting, and puts our discovery in context.

*It would be helpful to start off the Results section with an introductory figure (there are no figure limitations in* eLife*) showing a picture of courting male and female flies together with complete series as in*
Figure 5*, but with parts expanded a bit like*
Figure 1*, of duets (from a few different pairs) produced during the courtship interaction. The authors can use this introductory figure to describe the features of the signals that they use to define pulses and inter-pulse intervals, bouts etc. This figure should include a sequence where male and female song overlap if indeed they do under normal circumstances*.

We have included a new Figure 1, as the reviewers suggested. Former panels Figure 1 are now included in this new Figure 1 (as new panels D-F). The remaining panels from the former Figure 1 have been moved to a new Figure 2. In panels A and B of the new Figure 1 we include a schematic of the courtship pair for male (A) and female (B) song production. In panel C of the new Figure 1 we plot examples of recorded song from 5 different male/female pairs. As overlaps between male and female song were extremely rare during courtship (see below), we included only one example of an overlap (shaded in green).

*2) Overlap of Male and Female Song: There was some confusion about the issue of song overlap and how this was handled in the data analysis that the authors should clarify. For example in the playback experiments of*
Figure 4
*was the considerable overlap excluded (“Moreover, due to issues (in these experiments) with identifying female pulses that overlap with the male playback (see Methods), we also calculated female response times from the end of the male bout (*Figure 4
*inset).”)? How often did the songs overlap in the courting data set? In the courtship song sequence of*
Figure 5
*there appears to be no overlap at all. Is this correct and truly typical in the data set? If overlap occurs are the songs disentangled or is the data excluded. If overlap is excluded from the mutual singing experiments would not alternation be enforced by the analysis especially given the variable latency between male song and female ‘response’? Reference to any potential overlap should be made in the introductory figure and rationale for handling overlap in the analysis expanded*.

Overlaps between female and male song during courtship were extremely rare (population median = 0%). We now plot the % overlaps for all wild type (non-manipulated) male/female pairs in Figure 1—figure supplement 3. We also include this information in the text of the manuscript (Results section) and in the figure legend of Figure 1. We included these overlaps in all of the data analysis in the paper, with the only exception being the playback experiments (now Figure 5). Because the amplitude of the male playback stimulus was constant, but female song in this experiment was of variable intensity, any overlaps were obscured and difficult to score. However, we did try to score overlaps if they occurred, and any that we scored were included in the data analysis presented in the main panels of Figure 5. In the inset of Figure 5, however, we calculated the female response time from the end of the male playback bout, in order to avoid any issue introduced by our difficulty in scoring overlaps in this particular experiment. To be clear, there was no difficulty in scoring overlaps in data from male/female courtship pairs, only for the playback experiment. The inset in Figure 5 was necessary to plot (as the Reviewers correctly point out) because if there was a problem with scoring overlaps in this experiment, we would artificially see a peak at the end of the male bout in the response time curve for the playback expt. As we show, the female response time curve more closely (but not perfectly: these curves are still significantly different) resembles the randomized curves when we examine response times from the end of the male bout (the inset in Figure 5), and we therefore conclude that the playback has only a weak effect on her song timing. To strengthen this argument, we now include a new panel in this figure (Figure 5). We repeated the playback experiments with the addition of deafening the female. We see no difference in the response time curves for arista cut versus intact females (compare Figure 5 with Figure 5). Any relationship between female song and the playback stimulus must therefore indirectly be due to the effect of playback on the male. We have expanded the explanation of these results in the text of the manuscript, accordingly (please see the subsection headed “*Drosophila virilis* females rely on non-acoustic cues to coordinate song timing with males”). We also have added text throughout the Results and Methods sections to explain how we treated overlaps throughout the paper.

*3) Sexually Monomorphic: The sexually dimorphic or monomorphic terminology may be difficult for the general reader to follow. The authors need to put this terminology more firmly in the context of standards in the literature and what is known about duetting in other species or consider alternatives. A particular type of song may be elicited by tapping on the abdomen, and this song can be produced by females or males. Thus the authors might consider using female-typical for this sort of song and sexually differentiated to deal with the issue that females don't produce male-type song patterns*.

We appreciate that the terms dimorphic and monomorphic are a bit ‘jargony’, so we now define them clearly in the text of the manuscript. We also supplement these terms (or replace them) with the suggested terminology of “female-like” “male-like” or “sexually differentiated”. We decided not to remove the terms completely because they are widely used in both the acoustic duetting and *Drosophila* courtship literature.

4) Other Concerns:

A) Male singing seems to be important for courtship success. Tapping may also important for courtship success, but is duetting itself? Does the total amount of the song matter or the timing? Is there a way to leave tapping intact but de-synchronize the song?

These are excellent questions we considered; however, our study focused more on the mechanisms underlying the coordination of acoustic signals rather than the ethological relevance of male vs. female songs in mating. To address these questions it would be ideal to, as the reviewers suggest, uncouple song timing from male tapping. We did this with the headless female experiments (there she receives only the tactile cues); however, these females lack a brain and never become receptive. The only sensory manipulation in our study (of all combinations tried) that disrupted duetting but left song production rates intact was removal of the arista of males (see Figure 2). We observed no change in copulation rates for arista-cut males, but this may be due to the fact (as we suggest in the text) that these males can still sense female wing spreading (which is a receptivity cue). Male contact with her genitalia and abdomen should not change in this manipulation, which therefore suggests that the tactile cues alone are not only sufficient to drive female song production (as we show with the headless female experiments), but they also are sufficient to make the female receptive. Tarsiless males produce almost no song (see Figure 4). Males therefore also need information collected through contact with the female abdomen to drive his song production circuits (thus, we could not examine the impact of song in the absence of tapping; even in the playback experiments, females required interaction with a wingless male for song production). The experiment with arista-cut males also suggests that the male does not need to hear the duet to mate with the female. To clarify these points, we have added to the Discussion.

*B) Male performance is altered by deafening (*Figure 1*), but it isn't clear if he needs to hear his own song or the female's*.

In Figure 2—figure supplement 2 we show that arista-cut (AC) males do not change the overall distribution of their inter-bout intervals. This argues that the reason the response time curves change in AC males (Figure 2) is because he needs to detect the female song to couple with her, not because deafening causes an overall change in his song structure. In the text we say: “This effect was specific to response timing, because AC males maintained wild type levels of song production (Figure 4, manipulation 3), and showed no change in inter-bout interval (IBI) or pulse frequency (Figure 2—figure supplement 2). Thus, males rely on hearing their partner’s song for acoustic coordination.”

*C) The authors might want to cite Peter Narins' studies on visual/auditory integration in tungara frogs as another example of multimodal integration. Peter described the phenomenon as “binding” an idea worth considering in terms of the value of multiple sensory inputs*.

Thank you for drawing our attention to this study. We have now referenced it in the Discussion section.

*D) The authors seem to assume that duetting is derived rather than ancestral. For those readers not au courant with Drosophilid phylogeny, it would be helpful to explain the basis of this assumption. Presumably it reflects the prevalence of song in males alone but is enough known? The observation that males sing and females listen can obscure the more general finding that sexual dimorphisms are lost more often than they are gained (*[65]*).*

We did some more reading (along with the Wiens review you recommended) and discovered that, for birds, the most recent studies suggest that female song production is in fact ancestral. We now include this statement and relevant references in the Introduction.

*E) The authors state that competition is a possible driving force for the emergence of female songs and propose experiments using two males and a female*. *Might not the competition be between females?*

This is an excellent point that we had not considered, and we now make it in the Discussion.

*F) The authors indicate that one of the promises of this system is to dissect the genes and neural circuitry in* virilis *(using clues from melanogaster) that control multimodal integration involved in duetting. The authors might expand and be specific about why/how this system will allow forward progress on this interesting topic*.

Thank you for this suggestion. However, after some discussion among the authors, we felt the lengthy (and heavily referenced) first paragraph of the Discussion already highlighted the potential of this system for addressing the genetic and neural circuit mechanisms underlying duetting (given the homologies between *melanogaster* and *virilis*, the available annotated genome sequence for *virilis*, and what is already known about courtship behavior and circuits in *melanogaster*). We therefore chose not to expand further on this section because the genetic/neural circuit tools do not currently exist in *virilis* to do the proposed experiments. In other words, we did not want to run the risk of over-stating the value of this system.